# Serum Uric Acid but Not Ferritin Level Is Associated with Hepatic Fibrosis in Lean Subjects with Metabolic Dysfunction-Associated Fatty Liver Disease: A Community-Based Study

**DOI:** 10.3390/jpm12122009

**Published:** 2022-12-03

**Authors:** Cheng-Han Xie, Li-Wei Chen, Chih-Lang Lin, Ching-Chih Hu, Cheng-Hung Chien

**Affiliations:** 1Department of Gastroenterology and Hepatology, Chang-Gung Memorial Hospital and University, Keelung 204, Taiwan; 2Community Medicine Research Center, Chang-Gung Memorial Hospital and University, Keelung 204, Taiwan

**Keywords:** lean, metabolic dysfunction-associated fatty liver disease, liver fibrosis, serum uric acid, serum ferritin, liver steatosis, nonalcoholic fatty liver disease

## Abstract

Elevated serum ferritin and uric acid levels are common in patients with fatty liver disease. This study assessed the association between serum ferritin and uric acid levels and liver fibrosis in subjects with lean metabolic dysfunction-associated fatty liver disease (MAFLD). This cross-sectional study used data from a community screening examination for metabolic syndrome from December 2018 to September 2019 at Keelung Chang Gung Memorial Hospital. Subjects with lean MAFLD were defined as those with a body mass index (BMI) < 23 kg/m^2^ and hepatic steatosis according to the MAFLD criteria. A total of 182 lean subjects were included and were divided into lean MAFLD and lean healthy groups. Serum ferritin and uric acid concentrations were positively correlated with liver fibrosis, regardless of whether FIB-4, APRI, or NFS were used as references. Univariate logistic regression analysis showed that age and uric acid were associated with advanced liver fibrosis. After adjusting for potential confounders, only uric acid level was statistically significant in predicting the advanced liver fibrosis (OR = 6.907 (1.111–42.94), *p* = 0.038) in the lean MAFLD group. We found that an elevated serum uric acid level is an independent factor associated with advanced liver fibrosis in lean MAFLD subjects by noninvasive fibrosis scores.

## 1. Introduction

Nonalcoholic fatty liver disease (NAFLD) has emerged as the leading cause of chronic liver disease, particularly in the Middle East and western countries. The global prevalence of NAFLD is estimated at 20–30% [1,2]. NAFLD is closely linked to obesity and metabolic syndrome and is strongly associated with insulin resistance (IR) [3]. Therefore, NAFLD has been considered as the hepatic manifestation of metabolic syndrome [4]. NAFLD encompasses a broad spectrum of diseases ranging from simple hepatic steatosis to nonalcoholic steatohepatitis (NASH), which can progress to cardiovascular diseases, cirrhosis, hepatocellular carcinoma, and terminal liver failure [5,6,7]. Although NAFLD is usually associated with obesity, the subphenotype of lean subjects also presents with NAFLD and is becoming increasingly prevalent. The prevalence of lean NAFLD subjects was reported as 10.2% and appeared to be more common in Asia [8]. Another meta-analysis suggested that approximately 40% of the global NAFLD population was classified as nonobese and almost one fifth was lean [9]. In recent studies, the lean NAFLD subjects presented more severe histological presentations compared with nonlean NAFLD subjects [10].

In 2020, the international consensus panel proposed a new nomenclature from NAFLD to metabolic dysfunction-associated fatty liver disease (MAFLD), which is a positive criteria that focuses on metabolic abnormalities regardless of alcohol consumption or other concomitant liver diseases [11,12].

It is crucial to accurately evaluate the severity of liver histology, diagnose early, and carry out regular follow-ups of MAFLD, as it has become a leading cause of chronic liver disease. A routine liver biopsy for MAFLD is not feasible for a large number of MAFLD subjects, but remains the gold standard for uncertain cases. Simple and inexpensive noninvasive methods involving inexpensive laboratory biomarkers to measure liver fibrosis are alternative methods for the same purpose.

Among the different serum biomarkers, serum ferritin and uric acid levels have emerged as possible predictors for evaluating the severity of liver injury in NAFLD. Serum uric acid and serum ferritin share possible common pathogenic mechanisms, in particular oxidative stress and IR [13]. Previous studies have shown that increased serum ferritin or uric acid levels are associated with a more severe liver histology in NAFLD patients, but some studies have shown conflicting results [14,15,16,17,18,19,20]. The association of liver fibrosis, MAFLD, and hyperuricemia has been explored recently. A retrospective cross-sectional study in Taiwan showed that hyperuricemia increased the associated risk of significant liver fibrosis [21]. Another longitudinal study in China found a bidirectional relationship between hyperuricemia and MAFLD [22]. However, studies exploring the association between serum ferritin and uric acid levels and hepatic fibrosis in lean MAFLD subjects are scarce. Thus, we assessed the noninvasive biomarkers that could potentially predict hepatic fibrosis in lean MAFLD subjects.

## 2. Materials and Methods

### 2.1. Study Design and Population

This community-based, cross-sectional retrospective study was conducted from December 2018 to September 2019. A total of 1107 subjects aged ≥ 20 years who lived in the northeastern part of Taiwan participated in the health examination. The subjects’ anthropometric measurements, demographic data, blood biochemistry, and abdominal ultrasonography were collected at the Chang Gung Memorial Hospital at Keelung Branch. The history of viral hepatitis and alcohol consumption of the patients, and their medical records, were obtained using interview questions. All procedures in the study were conducted in accordance with ethical principles. The Institutional Review Board of Chang Gung Memorial Hospital approved this study (IRB no.: 103-3886C). All participants agreed to the study conditions and signed an informed consent form before enrollment in the study. Informed written consent was obtained from all subjects.

### 2.2. The Definition of Lean MAFLD, Lean Health

A diagnosis of MAFLD was made based on steatosis and the presence of any one of the following three conditions: overweight/obesity (defined as body mass index, BMI ≥ 23 kg/m^2^), presence of type 2 diabetes mellitus, or lean (BMI < 23 kg/m^2^) with evidence of metabolic dysregulation [11,12]. Metabolic dysregulation was defined as the presence of two or more of the following seven conditions: (1) waist circumference ≥ 90/80 cm in Asian men and women; (2) blood pressure ≥ 130/85 mmHg or specific drug treatment; (3) plasma triglycerides ≥ 150 mg/dL or specific drug treatment; (4) plasma HDL-cholesterol < 40 mg/dL for men and < 50 mg/dL for women or specific drug treatment; (5) prediabetes (fasting blood glucose levels within 100 to 125 mg/dL, or 2 h postload glucose levels from 140 to 199 mg/dL or HbA1c from 5.7% to 6.4%); (6) plasma high-sensitivity C-reactive protein (hs-CRP) level > 2 mg/L; and (7) homeostasis model assessment (HOMA)-IR score ≥ 2.5. Lean healthy subjects were defined as those with BMI < 23 kg/m^2^, and with no evidence of hepatic steatosis or metabolic dysregulation.

### 2.3. Physical Examination and Biochemical Measurements

A physical examination to measure body temperature, heart rate, respiratory rate, blood pressure, body weight, height, and waist circumference was performed. Subjects were considered to have hypertension if their blood pressure was ≥140/90 mmHg or if they were taking antihypertensive drugs. BMI was calculated as the weight in kilograms divided by the height in meters squared (kg/m^2^). All subjects were asked to fast overnight for 8 h before blood samples were drawn. Venous blood sample tests were used to measure the complete blood cell count and differential count, high-sensitivity CRP (HS-CRP, normal range: <1.0 mg/L), aspartate aminotransferase (AST, normal range: male ≤ 40 U/L; female ≤ 32 U/L), alanine aminotransferase (ALT, normal range: male ≤ 50 U/L; female ≤ 35 U/L), alkaline phosphatase (ALP, normal range: 40–140 U/L), gamma-glutamyl transpeptidase (GGT, normal range: male: 8–61 U/L; female: 5–36 U/L), total bilirubin (normal range: ≤1.2 mg/dL), total cholesterol (normal range: <200 mg/dL), high-density lipoprotein (HDL, normal range: male ≥ 40 mg/dL; female ≥ 50 mg/dL), low-density lipoprotein (LDL, normal range: <100 mg/dL), triglyceride (TG, normal range: <150 mg/dL), fasting plasma glucose levels (normal range: 74–100 mg/dL), hemoglobin A1c (HbA1c, normal range: <5.7), insulin (normal range: 2.6–24.9 μU/mL), serum uric acid, and serum ferritin. The serum uric acid level was measured using the colorimetric method, and the serum ferritin level was measured using an electrochemiluminescence immunoassay method. The upper limit of the normal serum ferritin level for sex-specific dichotomous comparisons was set at >400 ng/mL in men and >150 ng/mL in women. The upper limit of the normal serum uric acid level was >7.0 mg/dL.

### 2.4. Metabolic Syndrome

Metabolic syndrome was diagnosed according to the National Cholesterol Education Program (NCEP) Adult Treatment Panel (ATP) III diagnostic criteria, if more than three of the following five metabolic criteria were present: elevated blood pressure, elevated fasting glucose, elevated triglycerides, reduced HDL, and increased waist circumference [23].

### 2.5. IR

The homeostatic model assessment for insulin resistance (HOMA-IR) index was calculated using the following formula: fasting plasma insulin (mU/L) × fasting plasma glucose (mmol/L)/22.5. IR was defined as an HOMA-IR > 2 [24].

### 2.6. Fatty Liver Iindex (FLI)

The FLI is a noninvasive method of assessing hepatic steatosis that combines four measures (body mass index (BMI), waist circumference, triglyceride, and GGT), calculated using the following formula:$$\mathrm{FLI}=\frac{e^{0.953\times \mathrm{loge}\left(\mathrm{triglycerides}\right)+0.139\times \mathrm{BMI}+0.718\times \mathrm{loge}\left(\mathrm{GGT}\right)+0.053\times \mathrm{waistcircumference}-15.745}}{1+e^{0.953\times \mathrm{loge}\left(\mathrm{triglycerides}\right)+0.139\times \mathrm{BMI}+0.718\times \mathrm{loge}\left(\mathrm{GGT}\right)+0.053\times \mathrm{waistcircumference}-15.745}}\;\times \;100$$

The cutoff value of FLI for hepatic steatosis was set at >15 [25].

Hepatic steatosis was diagnosed using abdominal ultrasonography or FLI > 15 [25].

### 2.7. Ultrasonography

Abdominal ultrasonography is a widely available, noninvasive imaging method that allows the accurate detection of fatty liver disease [26]. In this study, abdominal ultrasonography (Toshiba, Xario, Japan) was performed after fasting for at least 6 h to assess hepatic steatosis and its severity (graded as mild, moderate, severe) by experienced gastroenterologists who were blinded to the subjects’ laboratory values. The presence and severity of liver steatosis were based on the finding of increased echogenicity of liver–kidney contrast, deep attenuation, and an obscured intrahepatic vasculature or diaphragm [27].

A noninvasive assessment of hepatic fibrosis was performed with serologic tests as described below.

### 2.8. Fibrosis-4 (FIB-4)

The Fibrosis-4 (FIB-4) score is a simple and noninvasive method that was used to assess liver fibrosis in this study. 

The FIB-4 score was calculated using the following formula: $$\frac{\mathrm{Age}\;\left(\mathrm{years}\right)\times \mathrm{AST}\;\left(U/L\right)}{\mathrm{Platelets}\;\left(10^{9}/L\right)\times \sqrt{\mathrm{ALT}}\;\left(U/L\right)]}$$

An FIB-4 index C ≥ 2.67 had an 80% positive predictive value and an FIB-4 index < 1.30 had a 90% negative predictive value for predicting NASH with advanced liver fibrosis [26]. The cutoff value of FIB-4 for predicting advanced fibrosis was set at ≥2.67 [28].

### 2.9. AST to Platelet Ratio Index (APRI)

The APRI was calculated using the following formula: [(AST/upper limit of normal])/platelet counts (10^9^/L)] × 100. The cutoff APRI value for predicting advanced liver fibrosis was set at >1 [29].

### 2.10. NAFLD Fibrosis Score (NFS)

The NFS was calculated according to the following formula: −1.675 + 0.037 × age (years) + 0.094 × BMI (kg/m^2^) + 1.13 × IFG/diabetes (yes = 1, no = 0) + 0.99 × AST/ALT ratio − 0.013 × platelet (×109/L) − 0.66 × albumin (g/dL) [30]. A score lower than the low cutoff score (−1.455) may exclude advanced fibrosis (F0-2) (negative predictive value of 93%) and a score greater than the high cutoff score (0.676) may predict the presence of advanced fibrosis (F3-4) (positive predictive value of 90%). Scores of −1.455 and 0.676 were recognized as indeterminate.

### 2.11. Statistical Analyses

Continuous variables are expressed as the mean and standard deviation (SD), and categorical variables are expressed as numbers and percentages. Between-group comparisons of continuous variables were conducted using Mann–Whitney U tests. Categorical variables were analyzed using a chi-square test. Correlation analyses were performed using Pearson’s correlation coefficients. Simple and multivariate logistic regression analyses were used to evaluate the risk of developing steatosis and liver fibrosis. Multivariate logistic regression models with a forward stepwise method were used after adjusting for age, sex, and factors that were significant in the univariate analysis. The *p* values were 2-tailed and *p* < 0.05 was considered statistically significant. Statistical analyses were performed using SPSS version 26.0 (SPSS Inc., Chicago, IL, USA).

## 3. Results

### 3.1. Baseline Characteristics

Initially, 1107 subjects were enrolled in the study. Subjects with BMI < 23 kg/m^2^ were included (*n* = 365). According to the status of hepatic steatosis, lean subjects were divided into fatty liver (*n* = 99) and nonfatty liver (*n* = 266) groups. Finally, 182 subjects (lean MAFLD group and lean healthy group) were included in the analysis (Figure 1). 

### 3.2. Differences in Laboratory Tests between Two Lean Subjects

Lean MAFLD subjects were older and had a higher percentage of diabetes, metabolic syndrome, or hyperuricemia than lean healthy subjects (Table 1). Lean MAFLD subjects had more metabolic abnormalities (waist circumference, blood pressure, TG, HDL, fasting glucose, HbA1c, IR, HS-CRP, and fatty liver index), liver enzymes, inflammatory markers, and higher noninvasive hepatic fibrosis scores (*p* < 0.01).

Figure 2 shows a box plot of serum ferritin and uric acid levels for the two lean groups. As there were some discrepancies in serum ferritin levels between the study subjects, the box plot of serum ferritin was prepared using the logarithmic transformation. Lean MAFLD subjects tended to have higher serum ferritin and uric acid levels than lean healthy subjects.

Figure 3 shows the number of metabolic abnormalities in the lean MAFLD group. The prevalence of prediabetes in lean MAFLD subjects was 92% (45/49), followed by hypertension (57%, 28/49).

### 3.3. Correlations of Hepatic Fibrosis 

Table 2 shows the correlation between the serum ferritin and uric acid levels. Positive correlations were observed between serum ferritin and uric acid concentrations and liver fibrosis.

### 3.4. Associated Factors for Liver Fibrosis and Steatosis 

The results of the logistic regression analysis used to determine the significant predictors of the occurrence of liver fibrosis in this study are shown in Table 3. Univariate logistic regression analysis showed that age and high uric acid level were associated with advanced liver fibrosis. In the multivariate logistic regression analysis, only high uric acid levels were a statistically significant predictor of advanced liver fibrosis.

The logistic regression analysis parameters shown in Table 4 determined significant predictors for the occurrence of liver steatosis. Univariate logistic regression analysis revealed that HOMA-IR, TG, uric acid, and ferritin levels were associated with liver steatosis. In the multivariate logistic regression analysis, only high TG levels were significantly associated with liver steatosis after adjusting for confounding factors.

## 4. Discussion

In this study, we showed that the serum uric acid level was an independent predictor for evaluating advanced liver fibrosis in lean MAFLD subjects, whereas serum ferritin level was not. The lean MAFLD subjects were a distinctive group who had a normal BMI but excess visceral adiposity and IR, as well as metabolic dysfunction, the so-called metabolically obese normal-weight individuals [31]. Pathophysiological mechanisms in lean MAFLD are not totally understood and may include IR, altered body composition, genetic mutations, epigenetic changes occurring early in life, and a different pattern of gut microbiota [32]. In a cross-sectional study in Taiwan, lean NAFLD was found in 4.2% of non-obese subjects [33]. Our data indicated that 13.4% (49/365) of nonobese subjects had MAFLD. Additionally, both serum ferritin and uric acid levels were higher in the lean MAFLD group than in the lean healthy group. 

Common causes of hepatic macrovesicular steatosis include obesity, type 2 diabetes, IR, dyslipidemia, and metabolic syndrome. Persistent hepatic steatosis could progress hepatic injury and fibrosis [34]. The “two-hit hypothesis” plays an important role in NAFLD pathogenesis, and inflammation in adipocytes has been linked to IR [35]. In previous studies and meta-analyses, elevated serum ferritin and uric acid levels were independently associated with NAFLD [36,37,38,39]. In our study, the serum ferritin and uric acid levels were associated with liver steatosis in a univariate logistic regression but not in a multivariate logistic regression. This may be related to that fact that most of the lean MAFLD subjects in this study had low BMI (<23 kg/m^2^) and mild metabolic abnormalities, including hyperuricemia (14.3%), type 2 diabetes mellitus (20.4%), metabolic syndrome (30.6%), and hypertriglyceridemia (30.6%). The other factor was that liver steatosis was identified with abdominal ultrasonography or a fatty liver index score, which could lead to measurement errors.

Serum ferritin is an acute phase protein of inflammation and is nonspecifically elevated in a wide range of systemic inflammatory conditions such as obesity, diabetes mellitus, metabolic syndrome, and fatty liver disease [36,40,41]. Previous studies have demonstrated that the serum ferritin level is an independent predictor of advanced hepatic fibrosis [17,19,29,42]. The exact mechanism is not totally understood, but the ferritin may be intimately involved in many processes related to NASH pathogenesis, including IR, excessive intracellular fatty acids, inflammatory process, oxidant stress, and fibrogenesis. Ferritin is also closely linked with proinflammatory cytokines and oxidative stress [43], and is able to activate hepatic stellate cells involved in liver fibrosis [44]. Previous studies have examined serum ferritin as a noninvasive biomarker to evaluate hepatic fibrosis. Some studies have demonstrated a positive correlation with liver fibrosis, while others have shown opposite results. Bugianesi et al. indicated that IR is an independent risk factor for advanced fibrosis and that increased ferritin levels are markers of severe histologic damage [20]. Kowdley et al. showed that an elevated serum ferritin level >1.5 from the upper normal limit is associated with a worsened hepatic histology and advanced hepatic fibrosis [17]. In contrast, Angulo et al. found that serum ferritin alone has a low level of diagnostic accuracy for detecting liver fibrosis [18]. In our study, we demonstrated that the serum ferritin level was not associated with advanced liver fibrosis in lean MAFLD subjects using noninvasive fibrosis scores. This may be explained by the “two-hit hypothesis” [35], with low lipid deposition, less IR, low oxidative stress, and inflammatory cytokines in our lean MAFLD subjects. The characteristics of these subjects included a low BMI < 23 kg/m^2^ and about 70% of subjects had only mild liver steatosis.

Serum uric acid is the final product of purine metabolism in humans and elevated uric acid levels can be increased by high purine food or fructose intake. Serum uric acid is an important antioxidant in vitro and elevated uric acid levels may be a protective response capable of eliminating the harmful effects of free radical activity and oxidative stress [45]. However, uric acid is also one of the damage-associated molecular patterns (DAMP)-activated proinflammatory cytokines, similar to IL-1b, and IL-18 secretion and activated hepatic stellate cells induced liver fibrosis [46]. Hyperuricemia was found to be strongly associated with endothelial dysfunction, a reduction in endothelial nitric oxide, and predisposed subjects to develop IR and metabolic syndrome in an animal model [47,48]. The other associated mechanism includes the increased oxidative stress of endoplasmic reticulum, mitochondrial dysfunction, and NLRP3 inflammasome induced by uric acid [35,49]. It has been widely noticed that hyperuricemia increases the risk of gout, cardiovascular disease, diabetes mellitus, and NAFLD [37,50,51]. Petta et al. showed that hyperuricemia is independently associated with hepatocellular ballooning and lobular inflammation in NAFLD patients [14]. Afzali et al. found that the serum uric acid level was associated with the development of cirrhosis [16]. In our study, we demonstrated that the serum uric acid level was associated with advanced liver fibrosis using noninvasive fibrosis scores in lean MAFLD subjects, and this finding corresponded with previous studies. In most of the lean MAFLD subjects in our study with prediabetes, we hypothesized that IR and endothelial dysfunction may have played a vital role in the pathogenesis of NASH-related fibrosis. Serum uric acid and serum ferritin share possible similar pathogenic mechanisms, in particular oxidative stress, chronic inflammation, and IR [13]. Serum uric acid is an inexpensive noninvasive biomarker for evaluating advanced liver fibrosis and combining it with other scoring systems may help to improve its predictive power. 

## 5. Limitations

There were several limitations in our study. First, liver steatosis and fibrosis in our study were identified using abdominal ultrasonography and blood biomarker scores (FIB-4, ARFI, and NFS), as liver biopsy could not be performed in healthy individuals. Moreover, we did not perform FibroScan or share wave elastography (SWE) to detect liver fibrosis because abdominal ultrasonography was performed with a Toshiba Xario ultrasound machine, with which neither FibroScan nor SWE were available. Some noninvasive scores, such as FIB-4, ARFI, and NFS, were applied in the mass screening for hepatic fibrosis evaluation, although there may still have been some bias. Second, this was a community-based, cross-sectional study that included a small number of subjects without longitudinal follow-up, so its findings may not be generalizable to other populations. Third, there was an uneven gender distribution in this study. The female predominance (80%) in our participants was due to this being part of a community health examination. Most women among our participants were housewives and might have had more time for a health examination. Further large number and long-term studies using SWE, FibroScan, or liver biopsy to detect hepatic fibrosis are required to validate the findings of the current study. 

## 6. Conclusions

In conclusion, our study demonstrated that hyperferritinemia and hyperuricemia are common in lean subjects with MAFLD. However, only elevated serum uric acid levels were independently associated with advanced liver fibrosis in lean MAFLD subjects in the Keelung community, detected using noninvasive fibrosis scores. Although uric acid is not a routine test in clinical practice for patients with lean MAFLD, this study revealed that the uric acid level may be a potential biomarker for advanced liver cirrhosis in these subjects. Further investigative studies are required to clarify the underlying mechanisms of this finding and its clinical use.

## Figures and Tables

**Figure 1 jpm-12-02009-f001:**
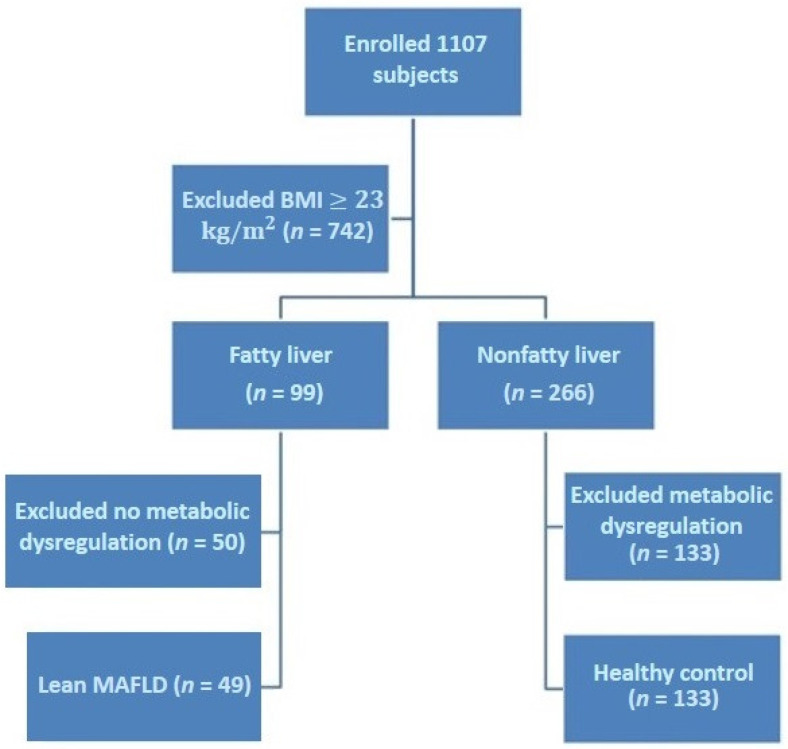
Flow chart of subject selection.

**Figure 2 jpm-12-02009-f002:**
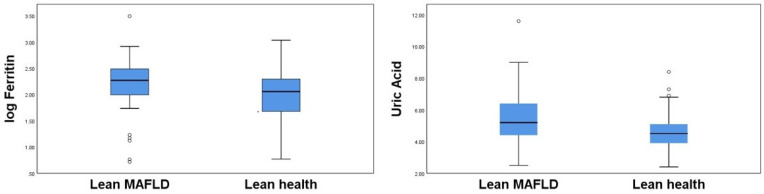
Box plot representation of ferritin and uric acid in lean MAFLD and healthy control group.

**Figure 3 jpm-12-02009-f003:**
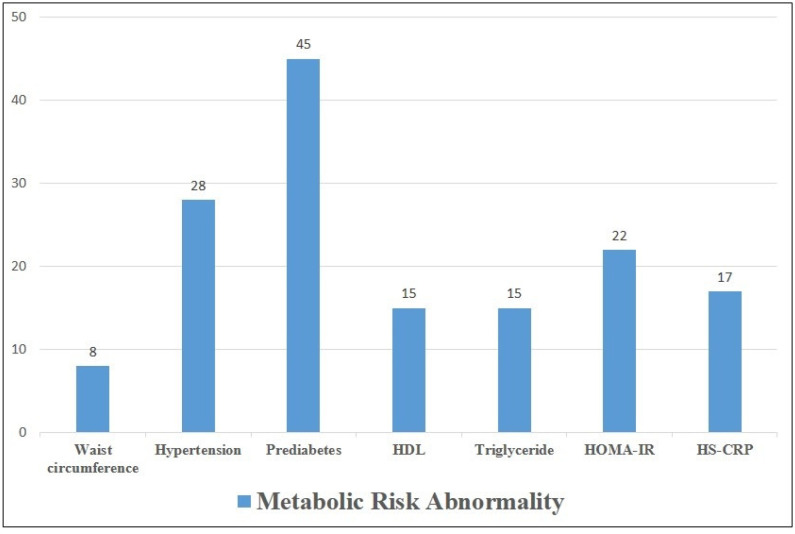
Number of metabolic risk abnormalities in lean MAFLD group.

**Table 1 jpm-12-02009-t001:** Baseline demographic and biochemical characteristics of study subjects.

	Total (*n* = 182)		
Variables	Lean MAFLD (*n* = 49)	Healthy Control (*n* = 133)	*p* Value
Demographics			
Male (*n*, %)	9 (18.4%)	25 (18.8%)	0.947
Age (years)	64.41 ± 9.29	51.06 ± 14.90	<0.001
Hypertension (*n*, %)	28 (57.1%)	0 (0%)	<0.001
Diabetes mellitus (*n*, %)	10 (20.4%)	0 (0%)	<0.001
Metabolic syndrome (*n*, %)	15 (30.6%)	0 (0%)	<0.001
Hyperuricemia (*n*, %)	7 (14.3%)	2 (1.5%)	<0.001
Hyperferritinemia (*n*, %)	26 (53.1%)	33 (24.8%)	<0.001
Hepatic steatosis			
Hepatitis steatosis (*n*, %)	49 (100%)	0 (0%)	<0.001
Mild	34 (69.4%)	0 (0%)	<0.001
Moderate	15 (30.6%)	0 (0%)	<0.001
Severe	0 (0%)	0 (0%)	<0.001
Fatty liver index	17.46 ± 17.17	5.24 ± 4.17	<0.001
Body measurement			
Height (cm)	154.95 ± 6.57	157.68 ± 7.71	0.029
Weight (kg)	51.60 ± 4.68	50.86 ± 6.71	0.230
BMI (kg/m^2^)	21.47 ± 1.12	20.41 ± 1.71	<0.001
Metabolic abnormality			
Waist circumference (cm)	75.48 ± 5.73	70.21 ± 6.27	<0.001
Systolic blood pressure (mmHg)	131.16 ± 14.83	114.00 ± 10.17	<0.001
Diastolic blood pressure(mmHg)	77.12 ± 9.51	69.20 ± 7.25	<0.001
Total Cholesterol (mg/dL)	190.7 ± 39.14	195.79 ± 31.98	0.377
Triglyceride (mg/dL)	140.02 ± 92.37	86.76 ± 54.02	<0.001
LDL (mg/dL)	122.49 ± 8.03	123.17 ± 32.42	0.786
HDL (mg/dL)	60.90 ± 21.40	70.14 ± 15.37	<0.001
Glucose AC (mg/dL)	106.96 ± 21.17	90.19 ± 7.07	<0.001
Insulin (μU/mL)	12.49 ± 15.93	5.88 ± 2.74	<0.001
HbA1c (%)	6.06 ± 0.54	5.46 ± 0.30	<0.001
HOMA-IR	3.86 ± 7.98	1.32 ± 0.64	<0.001
Liver bioenzyme			
AST (U/L)	34.92 ± 64.12	21.45 ± 8.96	0.003
ALT (U/L)	29.14 ± 23.61	22.06 ± 11.72	0.015
ALP (U/L)	80.57 ± 21.12	68.05 ± 19.99	0.001
Total bilirubin (mg/dL)	0.61 ± 0.36	0.69 ± 0.34	0.014
γ-GT (U/L)	48.29 ± 110.90	18.63 ± 26.76	<0.001
Inflammatory marker			
WBC (10^3^/uL)	5.87 ± 2.08	5.62 ± 1.94	0.611
Uric Acid (mg/dL)	5.40 ± 1.73	4.54 ± 1.05	0.001
Ferritin (ng/mL)	262.83 ± 442.30	143.68 ± 145.46	0.001
HS-CRP (mg/L)	4.35 ± 6.30	0.87 ± 0.76	<0.001
Hepatic fibrosis scores			
FIB-4 score	1.87 ± 2.85	1.03 ± 0.88	<0.001
NFS	−2.31 ± 1.65	−3.43 ± 1.30	<0.001
APRI score	0.43 ± 1.04	0.22 ± 0.15	0.002

Data are presented as the mean ± SD when appropriate (95% CI) or number (percentage). BMI, body mass index; HDL, high-density lipoprotein; LDL, low-density lipoprotein; HbA1c, hemoglobin A1C. HOMA-IR, homeostatic model assessment of insulin resistance; HS-CRP, high-sensitivity C-reactive protein; AST, aspartate transaminase; ALT, aspartate transaminase; ALP, alkaline phosphatase; FIB-4 score, fibrosis-4 score; NFS, NAFLD Fibrosis Score; APRI score, AST to platelet ratio index score. Continuous variables were analyzed using a Mann–Whitney U test, and categorical variables were analyzed using a chi-square test.

**Table 2 jpm-12-02009-t002:** Correlation between serum ferritin, uric acid concentration, and lean MAFLD metabolic risk.

	Ferritin		Uric Acid	
Variables	Pearson Coefficient	*p* Value	Pearson Coefficient	*p* Value
Waist circumference (cm)	0.257	0.075	0.193	0.184
Systolic blood pressure (mmHg)	0.130	0.372	0.218	0.132
Diastolic blood pressure (mmHg)	0.061	0.677	−0.008	0.955
HbA1c (%)	−0.191	0.189	−0.150	0.304
HDL (mg/dL)	0.155	0.287	0.173	0.233
Triglyceride (mg/dL)	0.131	0.368	0.077	0.600
WBC (10^3^/μL)	0.006	0.966	0.140	0.337
HS-CRP (mg/L)	0.032	0.846	0.009	0.958
ALT (U/L)	0.664	<0.001	0.329	0.021
HOMA-IR	−0.072	0.622	−0.165	0.257
FIB−4 score	0.907	<0.001	0.635	<0.001
NFS	0.462	0.001	0.565	<0.001
APRI score	0.945	<0.001	0.567	<0.001

**Table 3 jpm-12-02009-t003:** Univariate and multivariate analysis of variables associated with risk of advanced liver fibrosis with FIB-4 > 2.67.

Variables	Univariate Analysis		Multivariate Analysis *	
	OR(95 CI%)	*p* Value	OR(95 CI%)	*p* Value
Age (years)	1.199 (1.040–1.383)	0.012	1.184 (0.929–1.508)	0.172
Systolic blood pressure (mmHg)	1.075 (0.981–1.179)	0.122		
HbA1c (%)	0.153 (0.006–4.190)	0.266		
HOMA-IR	0.683 (0.266–1.753)	0.428		
Triglyceride (mg/dL)	1.001 (0.990–1.011)	0.892		
ALT (U/L)	1.025 (0.995–1.055)	0.103		
Hemoglobin (g/dL)	1.363 (0.661–2.811)	0.402		
Uric acid(mg/dL)	6.332 (1.288–31.134)	0.023	6.907 (1.111–42.94)	0.038
Ferritin (ng/mL)	1.002 (0.998–1.006)	0.261		
ALP (U/L)	1.024 (0.978–1.072)	0.309		
HS-CRP (mg/L)	0.990 (0.832–1.179)	0.913		

* adjusted for age and sex.

**Table 4 jpm-12-02009-t004:** Univariate and multivariate analysis of variables associated with risk of liver steatosis with FLI > 15.

Variables	Univariate Analysis		Multivariate Analysis *	
	OR(95 CI%)	*p* Value	OR (95 CI%)	*p* Value
Age (years)	1.052 (0.983–1.126)	0.143		
Systolic blood pressure (mmHg)	0.977 (0.938–1.018)	0.267		
HbA1c (%)	2.548 (0.821–7.909)	0.106		
HOMA-IR	1.757 (1.091–2.829)	0.020	1.445 (0.764–2.734)	0.258
Triglyceride(mg/dL)	1.031 (1.014–1.049)	<0.001	1.054 (1.013–1.096)	0.009
HDL(mg/dL)	0.982 (0.952–1.012)	0.237		
Uric Acid(mg/dL)	1.743 (1.107–2.744)	0.016	1.086 (0.375–3.143)	0.878
Ferritin	1.006 (1.000–1.011)	0.036	1.010 (0.996–1.024)	0.166
HS-CRP (mg/L)	1.027 (0.929–1.136)	0.597		

* adjusted for age and sex.

## Data Availability

The data presented in this study are available on request from the corresponding author.

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
