# Peer review of "Serum Uric Acid but Not Ferritin Level Is Associated with Hepatic Fibrosis in Lean Subjects with Metabolic Dysfunction-Associated Fatty Liver Disease: A Community-Based Study"

_jpm, 2022, doi:10.3390/jpm12122009_

Round 1
Reviewer 1 Report
Summary of research:
Xie and co-authors reported here the results of a retrospective study in which they showed how common are hyperferritinemia and hyperuricemia in lean subjects with MAFLD, and how elevated uric acid levels were associated with liver fibrosis in this group of patients.
Comments to the Author:
In my opinion, there are some more points that need to be discussed further:
1) No mention regarding Fibroscan or Share Wave Elastrography has been done. This deserves at least a comment if those procedures have not been performed for fibrosis detection and grading and a comment also regarding why this is a limitation for the study;
2) Talking about uric acid levels been independently associated with “advanced liver histology” in MAFLD patients is misleading, because no one of the patients underwent liver biopsy and this is nowhere mentioned, not even in the limitations of the study;
3) No reference is cited regarding the definition of metabolic dysregulation (Materials and Methods, 2.2);
4) It’s not clearly stated which of the reported investigation (ultrasound, scores, etc.) has been used to categorize patients with fatty liver (Baseline characteristics)
5) This group of patients count globally only on 20.4% of male individuals, which it makes this study not very well representative, especially in terms of risk factors associated with metabolic diseases. This should be mentioned in the limitations of the study;
6) No definition of prediabetes has been provided.
Reviewer 2 Report
I thank the authors for their effort on the article titled "Serum Uric Acid but not Ferritin Level is Associated with He- 2 patic Fibrosis in Lean Subjects with Metabolic Dysfunction-as- 3 sociated Fatty Liver Disease: A Community-based Study"
But i do not find it sufficient for publication for the following reasons:
- it includes only a small number of index subjects in a specific community (although they stated that already in the title)
- the comparison of demogafic data makes no sense and it is not surprising that almost all parameters were significant when comared to healty group
- the presentation of the correlation data is very confusing
- the authors stated themselfs that the diagnosis of liver fibrosis was based on a very subjective manner (ultrasonography), although the examiners were blinded to the lab findings of the patients
- the discussion is very poor, mainly introduction actually (line 278 - 284 and 297-308)
Round 2
Reviewer 1 Report
Thanks for the clarifications and the application of recommended changes.
Author Response
Thank you very much for your kind constructive comments on our manuscript.
Reviewer 2 Report
The manuscipt has been improved, still some core flaws in this study make it unsuitable for publication
Author Response
Thank you very much for your kind constructive comments on our manuscript.
We discuss the association between serum ferritin/uric acid and hepatic fibrosis. We have revised the section of discussion and we recognize there are some limitations in our study and add in the text. (Page 8-10, Discussion and limitation). We condense, adjust our Table and omit some items. (Page 5, table 1) Some new references were cited to explain the association in our study.